# Effect of an Environment Friendly Heat and Relative Humidity Approach on γ-Aminobutyric Acid Accumulation in Different Highland Barley Cultivars

**DOI:** 10.3390/foods11050691

**Published:** 2022-02-25

**Authors:** Shanshan Wang, Sumei Zhou, Lili Wang, Xiaojiao Liu, Yuling Ma, Litao Tong, Yuhong Zhang, Fengzhong Wang

**Affiliations:** 1Institute of Food Science and Technology, Chinese Academy of Agricultural Sciences, Beijing 100193, China; wang.3368@163.com (S.W.); wanglili03@caas.cn (L.W.); yulingma123@163.com (Y.M.); tonglitao@caas.cn (L.T.); 2Institute of Food Science and Technology, Tibet Academy of Agricultural and Animal Husbandry Sciences, Lhasa 850000, China; Liuxiaojiaoguyu@163.com; 3School of Food and Health, Beijing Technology and Business University, Beijing 100037, China; zhousumei1001@163.com

**Keywords:** heat and relative humidity treatment (HRH), highland barley, γ-aminobutyric acid, GABA-shunt pathway, polyamine degradation pathway

## Abstract

In this study, heat and relative humidity (HRH) treatment was applied in highland barley for γ-aminobutyric acid (GABA) accumulation. Tibetan highland barley cultivars (25) were selected for comparison and analysis. HRH treatment could accumulate GABA in several hours with low moisture content and high temperature, and the grains were treated for 2.5 h at 65 °C in this study. The GABA content of processed grains under HRH optimal condition ranged from 26.91 to 76.28 mg·100 g^−1^, which was significantly higher than the initial content (12.78–43.00 mg·100 g^−1^). The highest GABA accumulation capacity was observed in two-row yellow cultivars (YT1), increasing from 36.52 to 76.28 mg·100 g^−1^. Correlation analysis showed that the accumulation of GABA after HRH treatment was positively and significantly (*p* < 0.05) correlated with the contents of protein (0.52), total free amino acids (0.68), threonine (0.53), serine (0.51), glutamate (0.69), glycine (0.49), alanine (0.46), cysteine (0.57), tyrosine (0.50), lysine (0.53), proline (0.40), and glutamate decarboxylase (GAD) activity (0.62), which were closely related to GABA-shunt pathway. The polyamines contents, diamine oxidase (DAO) and polyamine oxidase (PAO) activities, as the substrates and critical enzymes of polyamine degradation pathway, showed no significant correlation with GABA accumulation. The results suggested that the main pathway of GABA accumulation in highland barley under HRH treatment was GABA-shunt pathway.

## 1. Introduction

Highland barley (*Hordeum vulgare* L.) with naked caryopsis is a type of barley mainly distributed in Tibetan Plateau at an elevation of 1400–4700 m [1], and is a traditional staple food, occupying over 70% of the crop lands in Tibet [2]. Due to the richness of various types of nutrients and bioactive compounds, like dietary fiber, protein, β-glucan, and phenolic compounds, highland barley has attracted extensive attention on hyperlipidemia, diabetes, regulation of blood pressure, and anti-atherosclerosis effects [3,4,5].

γ-Aminobutyric acid (GABA), a nonprotein amino acid, is regarded as an important inhibitory neurotransmitter in the mammalian central nervous system, having diverse physiological functions [6], such as antihypertensive, hypoglycemic, anti-cancer, diuretic and, sedative [7]. GABA naturally exists in plant materials, with concentrations of GABA from traces up to μmol·g^−1^, depending on the plant species, development stage, growth environment and processing conditions [8,9]. The GABA content of Tibetan Plateau highland barley was 18–28 mg·100 g^−1^ [10], which was higher than rice germ (10.75–18.62 mg·100 g^−1^) [11], fava bean (0.30–20.60 mg·100 g^−1^) [12], soybean (2.99 mg·100 g^−1^) [13], and wheat (0.12 mg·100 g^−1^) [14]. A higher GABA accumulation of highland barley may be induced by extreme climatic conditions, such as intense UV radiation, drought, and hypoxia [15]. In plants, GABA is primarily synthesized via the decarboxylation of L-glutamate, catalyzed by glutamate decarboxylase (GAD) [16]. This pathway is defined as GABA-shunt. Another contributor of GABA synthesis is the degradation of putrescine (Put), spermine (Spm), and spermidine (Spd), catalyzed by diamine oxidase (DAO) and polyamine (PAO) [17].

Previous studies showed that various abiotic and biotic stress can induce an elevation of the GABA level in plant seeds and tissue [18]. The conventional methods for GABA accumulation of cereals and pulses contain soaking and germination. Additionally, some co-processing based on soaking and germination prerequisites could also further promote GABA accumulation, such as ultrasonication, high hydrostatic pressure (HTP), and slightly acidic electrolyzed water [19,20,21]. However, these enrichment technologies of GABA involve long-time soaking and subsequent drying treatments, which are complex processes requiring enormous quantity of water and energy. The soaking pretreatment and germination make nutrients dissolve in the water, which not only results substantial nutritive value losses, but also increases the risk of microbial proliferation. Therefore, the environment friendly technology is essential to accumulate GABA.

The heat and relative humidity (HRH) treatment, first applied by Fukumori, which could accumulate GABA in several hours with low moisture content (16.0–18.5%) and high temperature (50–70 °C) [22]. The low moisture content of raw materials could reduce dying time and energy consumption, and prevent the taste deterioration and husk getting wrinkles. The capacity of HRH treatment on GABA accumulation has been confirmed in several cereals (rice, foxtail millet, and wheat buckwheat) and pluses (soybean, adzuki bean, mung bean, and mottled kidney bean) [22]. After HRH treatment, the GABA content of mung bean increased 7.52 times [23]. However, the mechanism of HRH treatment promoting GABA accumulation is not clear. Due to the hypoxia environment of Tibetan Plateau, the GABA content of highland barley is higher than other cereals and pluses [10]. The effect of HRH treatment on highland barley with high initial GABA content needs to be studied. In addition, variations in GABA accumulation have been reported in different brown rice and tomato cultivars upon stress conditions [11,12]. The influences of highland barley cultivar on GABA content and enrichment capacity are unknown.

In the present study, HRH treatment was used in highland barley to investigate the favorite conditions and the possible difference of GABA accumulation ability among cultivars. The relationship between GABA accumulation under HRH treatment and physical, chemical, and biochemical characteristics of highland barley was explored to confirm the contribution of GABA-shunt and polyamine degradation pathway.

## 2. Materials and Methods

### 2.1. Materials and Chemicals

Appendix A lists the twenty-five cultivars of highland barley that were used in this study: four two-rowed yellow highland barley (YT1-4), nine six-rowed yellow highland barley (SY1-9), and six-rowed purple hulless barley (PS1-12). They were planted at Lhasa, Tibet (altitude: 3658 m, oxygen content: 15.09%, mean temperature: 18.60 °C) in 2017. Whole grains were ground using a whirlwind mill (CT293, FOSS, Hillerod, Denmark) equipped with a 0.5 mm screen. Grains and grain flour were stored in double-layer self-sealing bags at 4 °C.

γ-Aminobutyric acid (GABA), putrescine (Put), spermidine (Spd), spermine (Spd), amino acid standard solution, sodium acetate, β-mercaptoethanol polyvinyl pyrrolidone (PVP), and pyridoxal 5-phosphate (PLP) were from Sigma Aldrich (Saint Louis, MO, USA). The HPLC-grade acetonitrile and methanol were purchased from Fisher Scientific (Waltham, MA, USA). All other chemicals and reagent used in the experiments were analytical grade.

### 2.2. Heat and Relative Humidity (HRH) Treatment

Moisture content was adjusted to 18% by adding deionized water to highland barley samples (50 g, wet basis). The samples were placed in a plastic tube, then placed on a shaker to make sure the even absorption of water. The samples were placed in a constant temperature and humidity box (KW-TH-49T, Dongguan KOWIN Testing Equipment Co., Ltd., Dongguan, China, temperature fluctuation ±0.2 °C, humidity 95 ± 1%) and treated for 2.5 h at 65 °C under the temperature set in the experiment (Appendix A). After HRH treatment, the highland barley samples were placed into a freeze dryer and dried to 10%. Finally, the samples were sealed using double-layer self-sealing and stored at 4 °C.

### 2.3. Determination of Germination Percentage, Kernel Weight, Starch and Protein

The germination percentage of 25 highland barley cultivars were determined based on International Seed Testing Association (ISTA) method. Kernel weight was the weight of one thousand highland barley seeds, which was used for analyzing the differences in grain average mass between cultivars [24]. The content of starch was determined using a Total Starch Assay Kit according to the enzymatic protocols from Megazyme. Protein was determined based on approved AACC method 46-12 (Kjeldahl method) [25].

### 2.4. Determination of GABA

The extract method of a slight modification was used [26]. Briefly a sample (0.25 g, dry basis) was extracted using a mixture of 70% ethanol (2.5 mL) by shaking it at room temperature for 30 min. The supernatant was obtained by centrifugation of the extract at 10,000× *g* at 4 °C for 10 min. This step was repeated three times, then combined the supernatant. The supernatant was filtrated through a 0.2 μm filter before GABA analysis.

The GABA content in highland barley were determined using HPLC (Agilent 1260, Agilent Technologies Inc., Palo Alto, CA, USA) according to the Chinese agricultural industry standard NY/T2890-2016. The filtrated supernatant was derivatized using dansyl chloride. The injection volume was 10 μL, the flow rate was 1.0 mL·min^−1^, and the temperature was set at 30 °C. Two mobile phases were 50 mM·L^−1^ sodium acetate (A) and acetonitrile (B). The elution gradient was that 0–20 min, 69% A; 20–30 min, 31% A; 30–35 min, 69% A. The chromatogram was detected at a wavelength of 436 nm. The GABA standard was analyzed as references under the same condition.

### 2.5. Determination of Free Amino Acid Composition

The samples (1 g, dry basis) were extracted using 0.1 M·L^−1^ HCl (15 mL) at room temperature with ultrasonic wave for 30 min [16]. The supernatant was obtained by centrifugation of the extract at 10,000× *g* for 10 min at 4 °C. A volume 1 mL of supernatant was added to a tube with 1 mL sulfosalicylic acid (10%) and placed into 4 °C for 60 min to settle protein, then centrifuged at 10,000× *g* for 10 min at 4 °C. The supernatant was used for determining the content of total free amino acid according to ninhydrin reaction. The extracts were filtered using a 0.2 μm filtration, then the composition of free amino acid was determined by an aim acid analyser (L-8800, Hitachi, Tokyo, Japan).

### 2.6. Determination of Polyamines

Polyamines were determined by HPLC using the method of Marek with a slight modification [27]. A sample (1 g, dry basis) was extracted using a mixture of 5% HClO_4_ (*v*/*v*) at ice bath for 1 h. The extract was centrifuged at 10,000× *g* for 20 min at 4 °C to get supernatant. The supernatant (1 mL) was mixed with NaOH (2 mL, 2 M·L^−1^), and then 20 μL benzoyl chloride/methanol (1/1, *v*/*v*) was added to the mixture. After incubation for 30 min at 37 °C, saturated NaCl was added to terminate benzoylation. Diethyl ether (3 mL) was added to extract benzoylated polyamines by shaking for 30 min and then centrifuged at 3000× *g* for 5 min. The upper etheric phase (1.5 mL) was evaporated to dryness at room temperature and dissolved in methanol (200 µL). The standards (Put, Spd and Spm) were prepared as the same operations. The benzoylated polyamines were filtered using a 0.2 μm filtration. The injection volume was 10 μL, the flow rate was 1.0 mL·min^−1^ and the temperature was set at 30 °C. Two mobile phases were water (A) and methanol (B). The isocratic elution condition was 36% A phase at a flow rate of 0.6 mL·min^−1^. The chromatogram was detected at a wavelength of 254 nm.

### 2.7. Assays of GAD, DAO and PAO

The GAD activity was determined using the method of Liu et al. [28] with minor modifications. A sample (1 g, dry basis) was smashed in liquid nitrogen, and then extracted using a mixture of potassium phosphate buffer (5 mL, 70 mM·L^−1^, pH 5.8) containing 2 mM·L^−1^ β-mercaptoethanol, 2 mM·L^−1^ EDTA and 0.2 mM·L^−1^ PLP at ice bath for 2 h. The extract was centrifuged at 10,000 rpm for 20 min at 4 °C. The supernatant was used for enzyme assay. The reaction mixture, which containing 200 µL of enzyme extract and 100 µL of substrate (1% Glu, pH 5.8), was incubated in a water bath at 40 °C for 2 h and then terminated in a boiling water bath for 5 min. The supernatant of the mixture was determined for GABA content. The release of 1 µM GABA per hour was defined as one unit (1 U) of GAD activity.

DAO and PAO activity were determined using the method of Liao with a slight modification [29]. A sample (1 g, dry basis) was smashed in liquid nitrogen, and then extracted using a mixture of potassium phosphate buffer (3 mL, 70 mM·L^−1^, pH 6.5) containing 5% PVP (*w*/*v*) at ice bath for 2 h. The extract was centrifuged at 10,000× *g* for 20 min at 4 °C. The supernatant was used for enzyme assay. Reaction mixtures (3.0 mL) contained crude enzyme extract (2.6 mL), peroxidase (250 U mL^−1^, 0.1 mL), and 4-aminoantipyrine/N, N-dimethylaniline (0.2 mL). The addition of Put or Spm/Spd (20 mM·L^−1^, 0.1 mL) was the priming of reaction. The 0.01 change of absorbance at 555 nm per minute was defined as one unit (1 U) of DAO or PAO activity.

### 2.8. Statistical Analysis

Experimental design was a completely randomized design. The experiments were performed in triplicate. All data were analyzed by one-way analysis of variance (ANOVA) and principal component analyses using SPSS statistical software (version 19.0) and the statistical significance was defined as *p* < 0.05. The data distribution and correlation analyses were performed using R statistical software (version 3.6.3).

## 3. Results and Discussion

### 3.1. Cultivar Variation in Germination Percentage, Kernel Weight, Starch and Protein

Seed germination, a prerequisite event for next-generation plant growth, is used in a variety of food products owing to the rise in metabolic activity [30,31]. Due to the importance of germination percentage for sprouting and improving metabolic activity, the germination percentages (GP) of 25 highland barley cultivars were determined [31]. The results showed the GP ranged from 47.33% to 96.67% (Figure 1a). There were 20 highland barley cultivars presenting a higher germination percentage than 75%. As a physical indicator of grain, 1000-kernel weight (KW) is used for analyzing the differences of grain average mass in varieties [24]. In this study, the KW of the highland barley ranged from 33.50 g to 54.65 g (Figure 1a), which was consistent with those reported by Annica [24]. The KW of six-rowed yellow highland barley was 49.26 ± 4.24 g, higher than two-rowed yellow and six-rowed purple cultivars (Figure 1b). The higher weight of six-rowed yellow cultivars might due to the more main cultivars planted in Tibet than local varieties.

The starch content of 25 highland barley cultivars ranged from 53.16% to 62.80% (Figure 1a), lower than the highland barley cultivars grown at high altitude (1200–3500 m) in India (56.30–68.00%) [32]. Protein contents varied from 9.77% to 18.18% (Figure 1a), higher than the hulless barley cultivars grown in India (9.58–12.2%) [33]. The average protein content was 12.55 ± 1.69%, which was consistent with the cultivars grown in south-central Qinghai-Tibet plateau (12.875 ± 0.6609%) [34]. The average protein content of two-rowed yellow cultivars (14.45 ± 2.84%) was higher than six-rowed yellow (11.67 ± 1.06%) and six-rowed purple cultivars (12.58 ± 1.14%) (*p* < 0.05) (Figure 1b). The difference could be ascribed to the genotype and environmental conditions [34].

### 3.2. Effects of Cultivars on GABA Content

The GABA contents of 25 highland barley cultivars were analyzed to screen out those cultivars, which may accumulate more GABA after HRH treatment. The results showed that there were significant differences in the initial GABA contents of 25 highland barley cultivars (Figure 2a). The initial GABA content was 12.78–43.00 mg·100 g^−1^, which was not low or even relatively high, compared with previous reports about other crops, such as rice germ (10.75–18.62 mg·100 g^−1^) [11], fava bean (0.30–20.60 mg·100 g^−1^) [12], soybean (2.99 mg·100 g^−1^) [13], or wheat (0.12 mg·100 g^−1^) [14]. The reasons for higher GABA content of highland barley than other cereals and pulses might be that the extreme environment of Tibet Plateau provides cold and anoxic stress on plants, which are good for GABA accumulation [35]. Among 25 highland barley cultivars, YT2 had the highest initial GABA content (43.00 ± 0.63 mg·100 g^−1^), while PS5 had the lowest content (12.78 ± 0.95 mg·100 g^−1^). The HRH treatment could promote GABA accumulation, but the increment presented prominent difference in multiple cultivars (Figure 2a). The GABA content of processed highland barley by HRH treatment was 26.91–76.28 mg·100 g^−1^, which was higher than untreated highland barley (12.78–43.00 mg·100 g^−1^) and other HRH processed cereals, such as brown rice (17.8 mg·100 g^−1^), white rice (16.5 mg·100 g^−1^), foxtail millet (24.1 mg·100 g^−1^), and wheat (18.0 mg·100 g^−1^) [22]. After HRH treatment, YT1 had the highest GABA content (76.28 ± 1.82 mg·100 g^−1^), while YT3 had the lowest content (26.91 ± 0.40 mg·100 g^−1^). The average initial content of GABA was similar in two-rowed yellow, six-rowed yellow, and six-rowed purple cultivars, whereas significant differences were observed after HRH treatment (Figure 2b). After treatment, the average GABA content of two-rowed yellow cultivars (50.47 ± 21.74 mg·100 g^−1^) was higher than six-rowed yellow (36.92 ± 4.55 mg·100 g^−1^) and six-rowed purple cultivars (37.82 ± 8.20 mg·100 g^−1^) (*p* < 0.05) (Figure 2b). These results were similar to previous reports, which showed that GABA accumulation was influenced by genotypes [12,36].

### 3.3. Cultivar Variation in Free Amino Acids Content and Composition

The contents of total free amino acids (FAA) ranged from 2.10–5.82 mg·g^−1^, which was similar to the results (0.69–6.29 mg·g^−1^) reported by Yoichi [37]. The highland barley presented a higher average content (2.94 ± 0.76 mg·g^−1^) of total free amino acids than barley grown in Japan (Figure 3a) [37]. There was no significant difference between two-row and six-row highland barley in the contents of total free amino acids, which was consistent with the previous studies [37]. The interrelationship between the GABA shunt and C/N metabolism facilitate the synthesis of amino acids, which were the substrates and products of GABA metabolism [38]. The results showed that the major contributors to the free amino acid pool were glutamate (Glu), aspartic acid (Asp), alanine (Ala), and arginine (Arg); similar results were confirmed by other research of barley, which showed that the major free amino acids were Glu, Asp, and asparagine (Asn) [37]. The average contents of Glu, Asp, Ala, and Arg were 34.66 ± 27.42 mg·100 g^−1^, 19.10 ± 5.63 mg·100 g^−1^, 11.75 ± 4.08 mg·100 g^−1^, and 15.66 ± 6.50 mg·100 g^−1^, respectively. The variable amplitude of Glu, Asp, Ala, and Arg were 9.97–138.12 mg·100 g^−1^, 11.17–34.77 mg·100 g^−1^, 6.23–22.82 mg·100 g^−1^, and 7.37–30.85 mg·100 g^−1^ (Figure 3b), respectively; similar results in barley were observed by Yoichi [37]. GABA could be synthesized from Glu, catalyzed by glutamate decarboxylase (GAD) [35]. The transamination of Ala was catalyzed by alanine aminotransferase (AlaAT), converting Ala to Glu [35]. The reversible conversion of Asp and Glu catalyzed by Asp aminotransferase (AspAT) also contributed to GABA production [35]. The higher contents of Glu, Ala, and Asp in highland barley than wheat and rice might be a reason for higher content of GABA [39,40]. The contents of Asp, Glu, threonine (Thr), serine (Ser), cysteine (Cys), leucine (Leu), phenylalanine (Phe), and histidine (His) of two-row highland barley were significantly higher than six-bow highland barley (data not list), while there was a significant difference in the contents of Leu and His between Yellow and purple highland barley (data not list).

### 3.4. Cultivar Variation in Polyamine Content and Composition

Spermine (Spm), spermidine (Spd), and putrescine (Put) are the most prevalent polyamines (PAs) in plants, and can be catalyzed by polyamine oxidases to synthesize GABA [35]. The contents of polyamines ranged from 1.12 to 6.30 mg·100 g^−1^, including Put (0.37–2.36 mg·100 g^−1^), Spm (0.38–2.07 mg·100 g^−1^), and Spd (0.13–1.87 mg·100 g^−1^) (Figure 4). The contents of Spm and Spd in barley reported by Srivastava were also in these range [37]. There were no significant differences in the three various types in the contents of PAs, Put, Spm, and Spd.

### 3.5. Cultivar Variation in GAD, DAO and PAO Activity

The synthesis of GABA including GABA shunt and polyamine degradation pathways, which were described as the decarboxylation of Glu catalyzed by GAD and degradation of PAs catalyzed by DAO and PAO [17,41]. In this study, the activity of GAD ranged from 0.18 to 10.99 U·g^−1^ (Figure 5a), the GAD activity of soybean reported by past research also in the range [13]. The highest GAD activity was observed in YT1. The GAD activity of two-row highland barley was significantly higher than six-row highland barley (Figure 5b). The ranges of DAO and PAO activity were 0.56–1.79 U·g^−1^ and 0.34–2.63 U·g^−1^ (Figure 5a), respectively. The DAO activity of two-row highland barley was significantly higher than six-row cultivars, but there was no significant difference in PAO activity between two-row and six-row highland barley (Figure 5b).

### 3.6. Correlations in Physical, Chemical and Biochemical Characteristics

In this study, the physical characteristic was kernel weight of grain, the biochemical characteristics including GP, GAD, DAO, and PAO activity, other indicators were chemical characteristics. There were significant and positive correlations (0.40 ≤ r ≤ 0.91) in free amino acids, except the significantly negative correlation between Arg and Asn (Figure 6). The remarkably positive correlations were observed in polyamines. Starch showed a negative and significant correlation with protein (r = −0.66) and FAA (r = −0.48) (Figure 6). There were no significant correlations in the GAD, DAO, and PAO activity. The prominent correlations were observed between biochemical and chemical characteristics. The GAD activity showed positive correlation with several free amino acids, but negative correlation with starch (r = −0.46). The remarkable and negative correlations were observed between free amino acids and DAO, PAO activity (Figure 6).

The initial GABA content (I-GABA) showed prominently positive correlations with GP (r = 0.54), KW (r = 0.46), Cys (r = 0.40), tyrosine (Tyr, r = 0.59), isoleucine (Ile, r = 0.50), Ans (r = 0.40), proline (Pro, r = 0.40), and GAD (r = 0.64) (Figure 6). The GABA content after HRH processing (P-GABA) showed significantly positive correlations with I-GABA (r = 0.85), the GABA accumulation content after HRH processing (A-GABA, r = 0.75), GP (r = 0.48), protein (r = 0.43), FAA (r = 0.56, Thr (r = 0.49), Ser (r = 0.41), Glu (r = 0.44), Cys (r = 0.59), Tyr (r = 0.68), Ile (r = 0.53), lysine (Lys, r = 0.44), Pro (r = 0.50), and GAD activity (r = 0.78) (Figure 6). The A-GABA showed remarkably positive correlation with protein (r = 0.52), FAA (r = 0.68), Thr (r = 0.53), Ser (r = 0.51), Glu (r = 0.69), glycine (Gly, r = 0.49), Ala (r = 0.46), Cys (r = 0.57), Tyr (r = 0.50), Lys (r = 0.53), Pro (r = 0.40), and GAD activity (r = 0.62) (Figure 6).

The main parameters related to GABA accumulation consisted of protein, free amino acids, and GAD activity, which were closely related to GABA-shunt. However, the substrate and enzymes of polyamine degradation pathway showed no significant correlation with GABA accumulation. The results suggested that the main pathway of GABA accumulation under the HRH treatment might be GABA-shunt pathway. Whereas the polyamine degradation pathway supplied 30% and 25% of GABA accumulation in germinated fava bean and tea, respectively, under hypoxia [29,31]. The practical contribution of polyamine degradation pathway to GABA accumulation needs to be further studied by inhibiting the activity of DAO and PAO to block up the polyamine degradation. The highest positive correlation with A-GABA was Glu, which was consistent with the results that the addition of Glu could promote GABA accumulation [26].

### 3.7. Principal Component Analysis (PCA)

Based on the similarities and differences of data, principal component analysis (PCA) could compress data and achieve dimensionality reduction of high dimension data. In order to explore the reasons for the difference of I-GABA, P-GABA, and A-GABA in various highland barley cultivars, the parameters that significantly correlated with I-GABA, P-GABA, and A-GABA were carried out with PCA. According to the greatest explanation of variation of the data, the principal components (PC) were chosen in this study.

For the parameters that significantly correlated with I-GABA, PCA explained 78.31% of the variation of the data in three components: PC1 (31.77%), PC2 (25.70%), and PC3 (20.84%) (Figure 7a). The variables Pro, GP, KW, Tyr, and Ile were the main formers of PC1(+). The variables Cys showed a high correlation with PC2(+) becoming its main formers. The variables Ans was the main contributor of PC3(+).

The loading plot indicated that the first three principal components account for 80.77% of the total variation that significantly correlated with P-GABA (PC1 = 40.41%, PC2 = 21.40%, PC3 = 18.96%) (Figure 7b). The most important contributors to PC1(+) were FAA, Thr, Ser, Glu, Ala, Cys, Ile, and Lys. The variables I-GABA, GP, Tyr, Pro, and GAD were the main formers of PC2(+). The PC3(+) represented A-GABA and protein.

For the parameters that significantly correlated with A-GABA, PCA explained 82.79% of the variation of the data in two components: PC1 (62.75%) and PC2 (20.04%) (Figure 7c). The important contributors to PC1(+) were FAA, Thr, Ser, Glu, Gly, Ala, Cys, and Lys, which were positioned close to each other indicating the high positive correlations between them. The GAD, Tyr, and Pro variables showed a high correlation with PC2(+).

### 3.8. Mechanism of GABA-Shunt Pathway for GABA Accumulation in Highland Barley under HRH Treatment

In this study, the accumulation of GABA after HRH treatment was significantly correlated with the initial content of protein, FAA, Thr, Ser, Glu, Gly, Ala, Cys, Tyr, Lys, Pro, and GAD activity. According to the results in Figure 6 and Figure 7 and those reported in previous studies on GABA-shunt and amino acid metabolism in plants [38,42], we proposed a pathway map of GABA synthesis to describe the main contributors to GABA accumulation in highland barley treated by HRH (Figure 8). As a precursor of GABA, Glu played an important role in GABA synthesis. Glu was the substrate of GABA synthesis, but also was the key core of amino acid metabolism. The GAD could catalyze the decarboxylation of Glu to GABA. The positive correlations of Glu content and GAD activity with GABA content were reported by the previous studies [28,43]. The transamination of Ala, Tyr, and Asp, catalyzed by aminotransferase (AT) converting Ala, Tyr, and Asp to Glu, also contributed to GABA production [35,42]. The conversion of Glu with Thr and Lys via the aspartate-family pathway also had effect on the GABA accumulation [38]. Upon stress, the Lys-ketoglutarate reductase (LKR) and saccharopine dehydrogenase (SDH) could convert Lys back into Glu [44]. In mitochondria, Pro could be degraded by Pro dehydrogenase (ProDH), and pyrroline-5-carboxylate dehydrogenase (P5CDH) to Glu [35]. In addition, the recent study showed that Pro also could be the precursor of GABA by a non-enzymatic reaction [45]. The conversion of Gly and Ser could be catalyzed by serine hydroxymethyltransferase (SHMT). At the same time, the Ser could contact with Glu via phosphoserine transaminase (PSAT) and phosphoserine phosphatase (PSPH) [43].

## 4. Conclusions

The GABA content of highland barley can be enhanced by HRH treatment conditions. The significantly different accumulations of GABA were observed in 25 highland barley cultivars. The accumulation of GABA after HRH treatment showed positive and significant correlation with protein, FAA, Thr, Ser, Glu, Gly, Ala, Cys, Tyr Lys, Pro, and GAD activity, which were closely related to GABA-shunt pathway. However, the substrate and enzymes of polyamine degradation pathway showed no significant correlation with GABA accumulation. The results suggested that the main pathway of GABA accumulation under the HRH treatment was GABA-shunt pathway.

## Figures and Tables

**Figure 1 foods-11-00691-f001:**
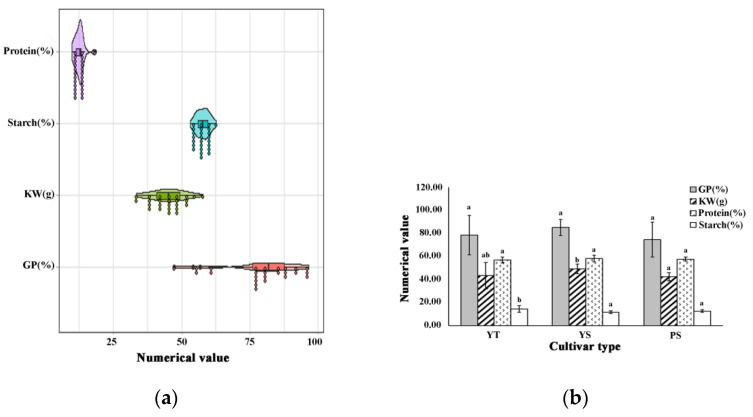
The cultivar variation in germination percentage, kernel weight, starch, and protein. (**a**) Cultivar variation in germination percentage, kernel weight, starch, and protein; (**b**) the difference of germination percentage, kernel weight, starch, and protein among different cultivar types. YT, two-rowed yellow highland barley; YS, six-rowed yellow highland barley; PS, six-rowed purple highland barley. GP, germination percentage; KW, kernel weight; Different letters denote significant difference among different highland barley cultivar types at the level *p* < 0.05.

**Figure 2 foods-11-00691-f002:**
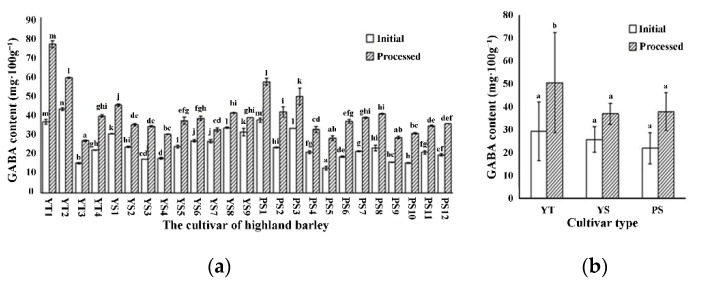
Effects of cultivars on GABA content. (**a**) The cultivar variation in GABA content; (**b**) the difference of GABA content among different cultivar types. Initial, the initial GABA content of highland barley; Processed, the GABA content of highland barley after HRH treatment. Different letters denote significant difference among different highland barley cultivars (**a**) or cultivar types (**b**) at the level *p* < 0.05.

**Figure 3 foods-11-00691-f003:**
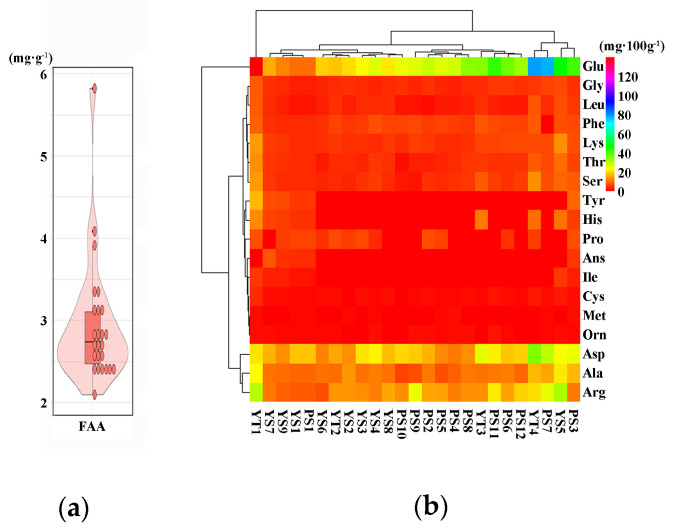
The cultivar variation in total free amino acids content (**a**) and amino acid composition (**b**).

**Figure 4 foods-11-00691-f004:**
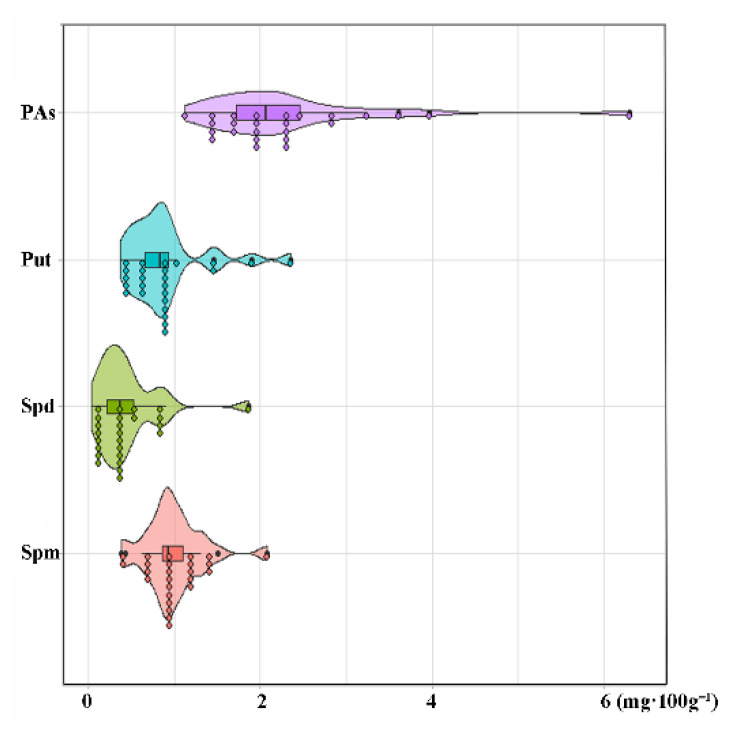
The cultivar variation in polyamine content and composition. PAs, total polyamines; Put, putrescine; Spm, spermine; Spd, spermidine.

**Figure 5 foods-11-00691-f005:**
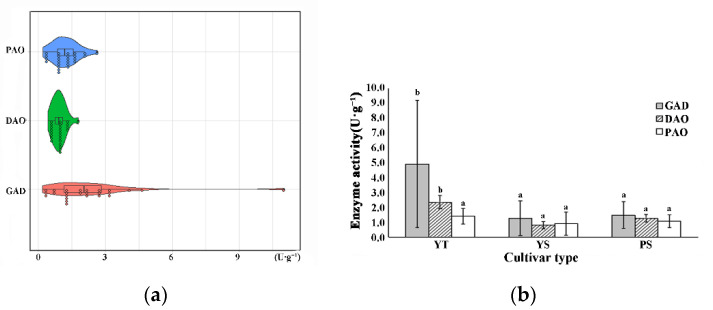
The cultivar variation in GAD, DAO, and PAO activity. (**a**) Cultivar variation in GAD, DAO, and PAO activity; (**b**) The difference of GAD, DAO, and PAO activity among different cultivar types. GAD, glutamate decarboxylase; DAO, diamine oxidase; PAO, polyamine oxidase. Different letters denote significant difference among different highland barley cultivar types at the level *p* < 0.05.

**Figure 6 foods-11-00691-f006:**
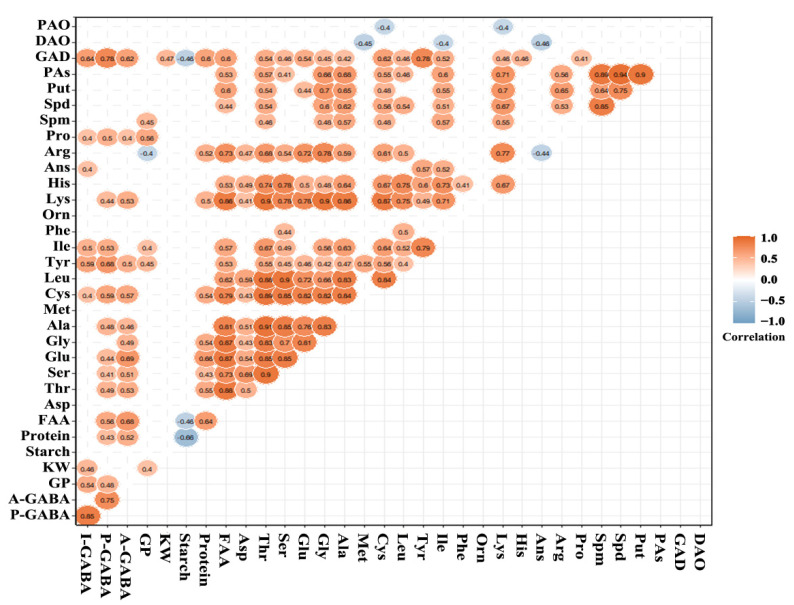
Correlations in physical, chemical and biochemical characteristics. The non-significant correlations were not displayed. I-GABA, the initial GABA content; P-GABA, the GABA content after HRH processing; A-GABA, the GABA accumulation content after HRH processing.

**Figure 7 foods-11-00691-f007:**
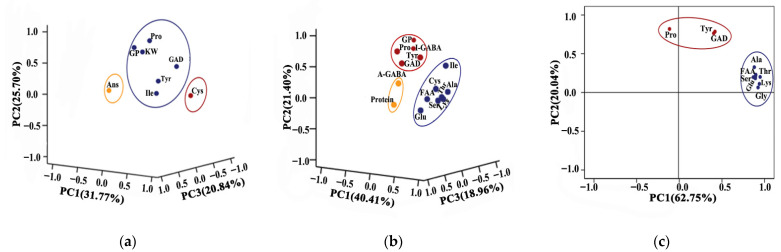
Principal component analysis. (**a**) Loading plot of PC1, PC2, and PC3, from parameters that significantly correlated with I-GABA; (**b**) loading plot of PC1, PC2, and PC3, from parameters that significantly correlated with P-GABA; and (**c**) loading plot of PC1 and PC2, from parameters that significantly correlated with A-GABA.

**Figure 8 foods-11-00691-f008:**
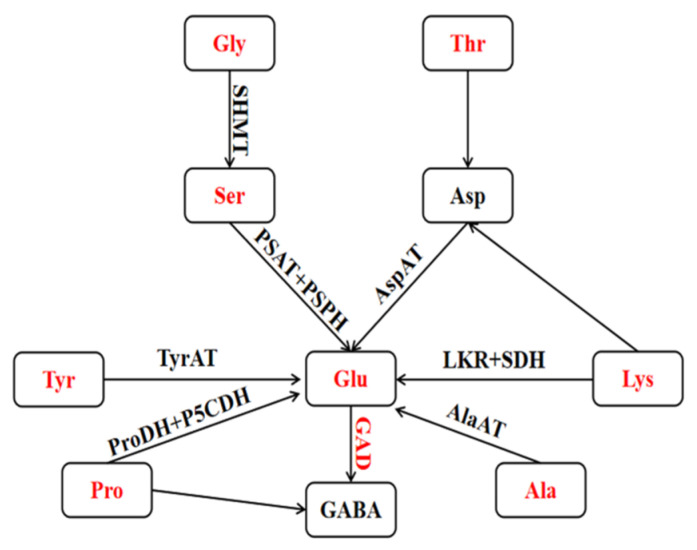
Mechanism of GABA-shunt pathway for GABA accumulation in highland barley under HRH treatment. The red marks represent the parameters, which were significantly and positively correlated with GABA accumulation under HRH treatment. GAD, glutamate decarboxylase; AlaAT, Ala aminotransferase; TyrAT, Tyr aminotransferase; AspAT, Asp aminotransferase; LKR, Lys-ketoglutarate reductase; SDH, saccharopine dehydrogenase; PrpDH, Pro dehydrogenase; P5CDH, pyrroline-5-carboxylate dehydrogenase; SHMT, serine hydroxymethyltransferase; PSAT, phosphoserine transaminase; PSPH, phosphoserine phosphatase.

## Data Availability

Data is contained within the article.

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
