# Peer review of "Effect of an Environment Friendly Heat and Relative Humidity Approach on γ-Aminobutyric Acid Accumulation in Different Highland Barley Cultivars"

_foods, 2022, doi:10.3390/foods11050691_

Round 1
Reviewer 1 Report
This manuscript entitled “Effect Mechanism of an Environment Friendly Heat and Relative Humidity Approach on γ-Aminobutyric Acid Accumulation in Different Highland Barley Cultivars” is an interesting and original study.
The paper is clearly presented and results are very useful. However, I have some suggestions:
- Figures can be improved.
- Figure 1b: Missing Y axis title and units.
- Indicate in the figure captions what are the letters that are on the columns (figures 1b, 2 and 5b).
- Figure 3: explain what the figure is…
- Figure 6: It is cut.
Reviewer 2 Report
I reviewed the manuscript entitled, Effect Mechanism of an Environment Friendly Heat and 3 Relative Humidity Approach on γ-Aminobutyric Acid 4 Accumulation in Different Highland Barley Cultivars. The manuscript is well written with appropriate scientific literature. Here are some suggestions/comments.
The title, effect mechanism…… is it effect of or mechanism of? I suggest effect of an environment…………….
Other comments and suggestion can be found below attached document
